# The Effects of Iron Deficiency on the Gut Microbiota in Women of Childbearing Age

**DOI:** 10.3390/nu15030691

**Published:** 2023-01-30

**Authors:** Hoonhee Seo, Seug Yun Yoon, Asad ul-Haq, Sujin Jo, Sukyung Kim, Md Abdur Rahim, Hyun-A Park, Fatemeh Ghorbanian, Min Jung Kim, Min-Young Lee, Kyoung Ha Kim, Namsu Lee, Jong-Ho Won, Ho-Yeon Song

**Affiliations:** 1Probiotics Microbiome Convergence Center, Soonchunhyang University, Asan-si 31538, Republic of Korea; 2Division of Hematology & Medical Oncology, Department of Internal Medicine, Soonchunhyang University Seoul Hospital, Seoul 04401, Republic of Korea; 3Department of Microbiology and Immunology, School of Medicine, Soonchunhyang University, Cheonan-si 31151, Republic of Korea

**Keywords:** iron-deficiency anemia (IDA), next-generation sequencing (NGS), gut microbiome, 16S rRNA gene-based metagenomics, dysbiosis, young women

## Abstract

Iron deficiency anemia (IDA) is the most prevalent and common nutritional deficiency worldwide and is a global health problem with significant risk, particularly among women of reproductive age. Oral iron supplementation is the most widely used and cost-effective treatment for iron deficiency and IDA. However, there are limitations regarding side effects such as enteritis, treatment compliance, and bioavailability. Intestinal microbiome characteristic research has been recently conducted to overcome these issues, but more is needed. Against this background, a metagenomics study on the 16S gene in the feces of young women vulnerable to IDA was conducted. As a result of analyzing 16 normal subjects and 15 IDA patients, significant differences in bacterial community distribution were identified. In particular, a significant decrease in *Faecalibacterium* was characteristic in IDA patients compared with normal subjects. Furthermore, in the case of patients who recovered from IDA following iron supplementation treatment, it was confirmed that *Faecalibacterium* significantly recovered to normal levels. However, no significance in beta diversity was seen compared with before treatment. There were also no differences in the beta diversity results between the recovered and normal subjects. Therefore, intestinal dysbiosis during the disease state was considered to be restored as IDA improved. Although the results were derived from a limited number of subjects and additional research is needed, the results of this study are expected to be the basis for developing treatment and prevention strategies based on host–microbiome crosstalk in IDA.

## 1. Introduction

Iron-deficiency anemia (IDA) is a global health problem and is considered the most prevalent and typical nutritional deficiency worldwide [1,2]. According to the Global Burden of Disease (GBD) report, IDA is one of the top five causes of Years Lived with Disability (YLD) and one of the ten most prevalent causes, with 1.24 billion cases [3,4]. Especially in developing countries, a low dietary intake has also been associated with iron supplementation and iron bioavailability, causing the disease to reach epidemic proportions [5].

IDA occurs when the synthesis of hemoglobin and iron-containing enzymes is restricted by iron, primarily due to malnutrition, iron deficiency in food, inadequate iron absorption, and increased iron absorption by the placenta or fetus [6,7,8]. Elemental iron involves in several fundamental biological processes, such as respiration, DNA replication, energy production, and cell proliferation [9]. A deficiency of this iron can lead to reduced physical strength, dyspnea, impaired thermoregulation, impaired immune function, and neurocognitive impairment [1]. Moreover, the IDA caused by this iron deficiency can exacerbate congestive heart failure or chronic kidney disease [10] and cause poor prenatal and neonatal health as well as delayed motor, cognitive, and growth development in children [11,12].

The percentage of iron deficiency is higher in women, and its risk is also higher, especially in women of childbearing age [7,13]. IDA was found to be the leading cause of elevated YLD in women in 35 countries [4] due to menstrual bleeding, pregnancy, and lactation [3,14,15]. In this regard, anemia in 2019 was 36.5% and 29.6% in pregnant and non-pregnant women, respectively [16]. IDA is prevalent in developing and developed nations, but it occurs to a greater extent among women in low- and middle-income nations, with 42.7% experiencing anemia during pregnancy [17]. In the United States, 5.0 ± 0.4% of pregnant women and 2.6 ± 0.7% of non-pregnant women were found to have IDA [3], similar to the figures in European countries [18,19].

Oral iron salt supplementation is the most popular and affordable method of treating iron deficiency and IDA [20]. However, it is is still limited by gastrointestinal side effects and noncompliance in up to 50% of patients, which is related to the nature of the intestinal flora and its associated bioavailability [21]. In particular, IDA is linked to dysbiosis along the gastrointestinal tract in the large intestine [22], and iron therapy also impacts the diversity and composition of the intestinal flora [23]. However, reports on host–microbiome crosstalk studies in IDA are still in their infancy.

Considering this research background, it is necessary to study the characteristics of the gut microbiome in IDA to obtain essential clues for improving IDA outcomes. Therefore, in this study, we focused on young women particularly vulnerable to IDA. Moreover, more studies have yet to be done on how iron supplementation affects human gut microbiota when treating IDA. Therefore, in this study, we also analyzed the fecal microbiome of normal subjects and IDA patients in young women and investigated subjects who recovered from IDA with iron supplementation.

## 2. Materials and Methods

### 2.1. Subjects Recruitment and Sample Collection

In this study, we recruited 31 premenopausal women between the ages of 20 and 50, including 15 IDA patients and 16 healthy subjects. The IDA patients were diagnosed following the WHO’s diagnostic criteria [24]. Control participants were self-reported to be healthy and had no disease at the time of recruitment. At the time of screening, only individuals with hemoglobin levels of more than 12 g/dL and ferritin levels of more than 15 ng/mL were included. Participants did not use antibiotics or other medical treatments affecting the gut microbiome during the 3 months before conducting the study. Stool sampling was performed according to the previously published manual [25]. IDA patients received oral (a tablet containing 80 mg of iron sulfate twice a day) or intravenous (the repeated administration of 200 mg of iron sucrose for a total of 1000 to 1500 mg) iron supplements for 3 to 6 months, and blood tests confirmed that their anemia improved. Then, sampling was performed again. The study was conducted according to the published ethical principles [26] and was approved by the Institutional Review Board (IRB no. 2020-08-017) of Soonchunhyang University Hospital. All the subjects signed a written consent before conducting this study.

### 2.2. DNA Extraction from Stool Samples

DNA was extracted from stool samples using the QIAamp DNA fast Stool Mini Kit (Qiagen, Hilden, Germany). Next, DNA concentration was measured using a Qubit-4 fluorometer (Thermo Fisher Scientific, Waltham, MA, USA). The quality of the extracted DNA was analyzed with electrophoresis on a 0.8% agarose gel. All DNA samples were stored at −20 °C until the following experiment was performed.

### 2.3. 16S rRNA Gene-Based Metagenomics

The V4 hypervariable region of the 16S rRNA gene was amplified from bacterial DNA extracted from feces. PCR was performed in a Veriti 96-well Thermal cycler (Applied Biosystems, Thermo Fisher, Waltham, MA, USA) according to the recommended Illumina amplicon PCR conditions. We separately used 5 µM of each of the primer sets, 10 ng of template DNA, the KAPA HiFi HotStart ReadyMix (Kapa Biosystems, Wilmington, MA, USA), and nuclease-free water in a final volume of 25 µL for each sample. Primer information is as follows. Forward primer: TCGTCGGCAGCGTCAGATGTGTATAAGAGACAG-CCTACGGGNGGCWGCAG; reverse primer: GTCTCGTGGGCTCGGAGATGTGTATAAGAGACAGGACTACHVGG-GTATCTAATCC. PCR products were then purified using AMPure beads (Beckman Coulter, High Wycombe, UK). Next, Nextera XT DNA Library Prep Kit (Illumina, CA, USA) was used to carry out the index PCR using the purified PCR product. Subsequently, indexed PCR products were purified, and all samples were diluted to 1 nM each in 10 mM of Tris, pH 8.5, and then 5 μL of each sample was taken and mixed by pooling them together. Pooled libraries (50 pMol) were mixed with 30% PhiX (Illumina, CA, USA) and sequenced on an iSeq 100 platform (Illumina, CA, USA).

### 2.4. Bioinformatics Analysis of 16S rRNA Gene Amplicons

Data were analyzed according to previously described procedures [27]. The EzBioCloud server (http://www.ezbiocloud.net, accessed on 15 December 2022) was used for data processing. Here, Trimmomatic (version 0.32) was utilized to evaluate the quality and and filtering of low-quality reads (<Q25), and Myers and Miller’s alignment algorithm was used for primer trimming [28]. Using the software programs HMMER and nhmmer (package version 3.2.1), samples without 16S rRNA encoding were detected [29]. In parallel, VSEARCH’s derep_full length command was applied to clusters of unique and duplicate reads [30]. EzBioCloud’s 16S rRNA database and VSEARCH were also applied for the taxonomic assignment [28,30,31]. Chimeric reads were filtered using UCHIME, and the cluster_fast command was utilized to identify sequences at lower taxonomic levels and generate operational taxonomic units (OTUs) [30,32]. Single-read OTUs were removed for further analysis.

In the alpha diversity analysis, species richness was calculated based on ACE, Chao1, Jackknife, and OTU [33,34,35], and species richness was analyzed using NPShannon, Shannon/Simpson, and phylogenetic diversity [36,37,38]. Beta diversity distance was analyzed based on the Jenson–Shannon, Bray–Curtis, Generalized UniFrac, and UniFrac methods, and PERMANOVA (Permutational Multivariate Analysis of Variance) was employed to assess the significance of the beta set between groups [39,40,41,42]. Taxonomic biomarker discovery was performed via statistical comparison algorithms of LEfSe (Linear discriminant analysis Effect Size) and Kruskal–Wallis H tests [43,44]. Functional profiles of samples were predicted based on KEGG (Kyoto Encyclopedia of Genes and Genomes) using PICRUSt (Phylogenetic Investigation of Communities by Reconstruction of Unobserved States) [45,46]. A p-value of less than 0.05 indicates the statistical significance of the data.

### 2.5. qRT-PCR Analysis

qRT-PCR analysis was conducted to quantify the genus *Faecalibacterium* and the species *Faecalibacterium prausnitzii* in each group of samples. The following primer sets were used in this study: HFB-F3 (GCTTTCAAACTGGTCG) and HFB-R5 (GAAGAGAAACGTATTTCTAC) specific for *Faecalibacterium* and FPR-2F (GGAGGAAGAAGGTCTTCGG) and Fprau645R (AATTCCGCCTACCTCTGCACT) specific for *F. prausnitzii* [47,48]. qRT-PCR was measured in triplicate using real-time PCR (CFX connect, BioRad, Hercules, CA, USA). Each well consisted of 1 μL of DNA (5 ng), 10 μL of a 2× Syber Green Master mix (BioRad, Hercules, CA, USA), 1 μL of each primer (2.5 μM), and 7 μL of qRT-PCR-grade water (Qiagen, Hilde, Germany). PCR results were analyzed using Graphpad Prism software (Ver. 8.0.263).

## 3. Results

### 3.1. Patient Profiles with Baseline Characteristics

The IDA group consisted of 15 patients diagnosed with IDA (serum ferritin < 20; hemoglobin < 12.0; MCB, mean corpuscular volume < 80) before treatment and without any other systemic disease. Of the ten patients whose samples were collected following therapy, eight were treated with oral iron and two were treated with intravenous iron due to the adverse effects of oral iron. The normal group consisted of 16 healthy premenopausal women. The age distribution, height, and weight of the two groups did not differ significantly (Table 1).

### 3.2. Averaged Taxonomic Composition

Differences in the mean taxonomic composition of the normal subjects, IDA patients, and those who recovered from IDA following iron supplementation were analyzed (Figure 1). The raw data for these results are shown in Appendix A, and only those with significant differences among taxa with a relative taxonomic abundance greater than 1% were extracted and are summarized in Table 2.

Differences between groups were identified in five classes (*Clostridia*, *Coriobacteriia*, *Negativicutes*, *Erysipelotrichi*, and *Betaproteobacteria*), three orders (*Clostridiales, Coriobacteriales*, and *Erysipelotrichales*), three families (*Ruminococcaceae, Coriobacteriaceae*, and *Erysipelotrichaceae*), and three genera (*Faecalibacterium, Collinsella*, and *Veillonella*). The *Clostridia* class, *Clostridiales* order, *Ruminococcaceae* family, and *Faecalibacterium* genus were same lineage and showed the most remarkable results, showing significant decreases in IDA patients compared with normal subjects. Moreover, in the samples recovered with iron supplementation, the level was again increased compared with the level of normal subjects, although not significantly. Normal subjects and recovered subjects did not significantly differ from one another. The other lineage, Coriobacteriia class, Coriobacteriales order, Coriobacteriaceae family, and Collinsell genus, had a lower composition ratio in IDA patients than normal subjects, but this was not significant. It was significantly lower in the recovered group compared to normal subjects. Regarding the Negativicutes class–Veillonella genus lineage, the Negativicutes class ratio was significantly higher in the IDA patients and the Veillonella genus ratio was significantly higher in the recovered group, respectively, compared with normal subjects. The Erysipelotrichi class–Erysipelotrichales order–Erysipelotrichaceae family lineage was confirmed to have a lower composition ratio, as it recovered after iron supplementation compared with before iron supplementation in IDA patients. The Betaproteobacteria class also showed similar results to those mentioned before. Additionally, those mentioned above were the only significant differences across all ranks and taxa.

### 3.3. Alpha Diversity Analysis

Alpha diversity was analyzed between normal subjects, IDA patients, and subjects who recovered from IDA following iron supplementation (Figure 2). Species richness was examined using Ace, Chao1, Jacknife, and OTU, and species diversity was examined using NPShannon, Shannon, Simpson, and Phylogenetic diversity. The three groups did not differ statistically significantly from one another as a result of any diversity outcomes.

### 3.4. Beta Diversity Analysis

Beta diversity was analyzed between normal subjects, IDA patients, and subjects who recovered from IDA following iron supplementation.

The beta set significance between groups was determined with PERMANOVA (Table 3). The applied Jenson–Shannon (*p* = 0.037), Bray–Curtis (*p* = 0.017), and Generalized UniFrac (*p* = 0.045)-based analysis showed significant differences between normal subjects and IDA patient groups, but the UniFrac-based analysis did not show any significant differences. However, no significant differences were seen in any analyses before and after iron supplementation in IDA patients. Additionally, subjects who recovered from IDA following iron supplementation showed no significant differences in any of the analyses compared with normal subjects.

A PCoA (principal coordinate analysis) analysis was conducted between groups of samples using four diversity metrics: Jensen-Shannon, Bray-Curtis, General-ized UniFrac, and UniFrac (Figure 3). As a result, it was confirmed that the distribution patterns of normal subjects and subjects who recovered from IDA following iron supplementation were similar. However, although it was not clearly differentiated, it was confirmed that the distribution pattern of the IDA patients was different than those of the normal subjects and the recovered subjects.

A clustering analysis was performed between groups using the unweighted pair group method with the arithmetic mean (UPGMA) method (Figure 4). The data showed that the normal group was distributed with a bias to the right, which was opposite to the distribution of the IDA patients. On the other hand, subjects who recovered from IDA following iron supplementation were evenly distributed and not biased to one side. 

### 3.5. Taxonomic Biomarker Discovery

Taxonomic biomarkers between groups were explored with LEfSe analysis and the Kruskal–Wallis H test (Figure 5). Only taxa with a LDA (linear discriminant analysis) score of 3 or higher, showing significant differences in the Kruskal–Wallis H test, were judged as potential biomarkers and are presented. Compared with normal subjects, the reduction in the *Clostridia* class–*Clostridiales* order–*Ruminococcaceae* family–*Faecalibacterium* genus lineage was found to be a key biomarker of IDA patients (Figure 5A). Moreover, in patients with IDA, a decrease in the *Erysipelotrichi* class–*Erysipelotrichales* order–*Erysipelotrichaceae* family lineage after iron supplementation is believed to be associated with recovery from IDA (Figure 5B). Here, an increase in the *Veillonella* genus and a decrease in the *Coriobacteriia* class–*Coriobacteriales* order–*Coriobacteriaceae* family–*Collinsell* genus lineage in recovered subjects compared with normal subjects were analyzed as biomarkers between the two groups (Figure 5C).

### 3.6. Prediction of Functional Biomarkers

An analysis of functional biomarker predictions was performed between groups of normal subjects, IDA patients, and subjects who recovered from IDA following iron supplementation (Table 4). Based on the LEfSe analysis, six orthologies (type IV secretion system protein VirD4, K03205; DNA replication protein DnaC, K02315; DNA topoisomerase III, K03169; chromosome partitioning protein, K03496; putative DNA primase/helicase, K06919; and DNA primase, K02316), four modules (uridine monophosphate biosynthesis, glutamine (+PRPP) => UMP, M00051; heme biosynthesis, glutamate => heme, M00121; glycogen biosynthesis, glucose-1P => glycogen/starch, M00854; beta-oxidation, acyl-CoA synthesis, M00086), and two pathways (lysosome, ko04142; insulin signaling pathway, ko04910) with a LDA effect size of greater than 2 were identified as functional biomarkers. ‘Heme biosynthesis, glutamine => heme’ module, ‘lysosome pathway’, and ‘insulin signaling pathway’ were higher in IDA patients than normal subjects; in subjects who recovered with iron supplementation, the values were again lower and similar to those of normal subjects. Except for those just mentioned, the functional biomarkers showed the opposite trend. In other words, the composition ratio was lower in IDA patients than in normal people. However, in the case of subjects who recovered from IDA following iron supplementation, the value tended to again increase by the same amount as normal subjects.

### 3.7. Quantification of Faecalibacterium Based on qRT-PCR

Quantitative analysis based on qRT-PCR was performed for the *Faecalibacterium* genus and *F. prausnitzii* in the feces of normal subjects, IDA patients, and subjects who recovered from IDA following iron supplementation (Figure 6). Data analysis showed that the *Faecalibacterium* load was lower in IDA patients than in normal samples and again significantly increased (*, *p* < 0.05) after iron supplementation treatment (Figure 6A). The amount of *Faecalibacterium* recovered following iron supplementation in the gut microbiome was also significantly different from that of the normal group samples (**, *p* < 0.01). Regarding F. prausnitzii, there was no significant difference between the normal and recovered groups. At the same time, it was significantly higher in the recov-ered groups when compared to IDA patients (**, *p* < 0.01) (Figure 6B). These PCR results confirm the NGS sequencing results, indicating the abun-dance of Faecalibacterium and F. prausnitzii in the samples. 

## 4. Discussion

As of 2019, the global incidence of anemia resulting from a lack of iron in the diet was higher than that of any other disease in all age groups except indi-viduals who were 95 years or older—specifically, in the case of prevalence per 100,000 people, it was higher in both males and females under the age of 9. The incidence of anemia caused by dietary iron deficiency was also higher in older males and younger females [49]. In terms of prevalent cases, it was high in children and adults, especially in women of childbearing age [49]. As such, studies on IDA-related intestinal microbial dysbiosis in infants between the ages of 6 to 34 months, with the highest occurrence of IDA, have already been reported [50]. Moreover, in this regard, survey results on iron fortification’s effect on children’s intestinal microflora have been reported [51]. Additionally, research on IDA and the microbiome in young women was recently published in December 2022, though it was limited to pregnant women [6]. However, although we investigated the literature, there has not yet been a report on IDA and the microbiome covering women of childbearing age who account for most of the prevalent cases of IDA. Accordingly, this study’s main objective was to examine the gut microbiome characteristics of normal subjects and IDA patients in women of childbearing age. Furthermore, the investigation was extended to subjects who had recovered from IDA with iron supplementation.

This study showed significant variations in the beta diversity of the distributional aspect of the bacterial community between normal subjects and IDA patients. Intestinal dysbiosis in IDA patients was also the same in the results of the studies on infants and pregnant women introduced above. In two studies targeting infants or pregnant women, the control group and IDA group comprised 10 patients each, and the beta diversity indices between groups were significant [6,50]. However, species diversity and richness, which are also measures of alpha diversity, did not significantly differ between the groups. As for the the specific pattern of change for each taxa, the *Clostridia* class, *Clostridiales* order, *Ruminococcaceae* family, and *Faecalibacterium* genus (all of which are of the same lineage) showed significant decreases in IDA patients compared with normal subjects. Clostridia decreased from 45.6% to 33.8%, *Clostridiales* decreased from 45.6% to 33.8%, *Ruminococcaceae* decreased from 31.2% to 19.6%, and *Faecalibacterium* decreased from 15.3% to 7.5%, showing a significant reduction range from 7.8% to 11.8%. In the case of other taxa showing significant differences among those with a composition ratio greater than 1%, the composition ratio was low in the 0.4% to 8.8% range. However, the range of increase and decrease was only between 0.2% to 4.3%. When focusing on the most specific genus among the ranks, there was a significant difference between *Collinsella* and *Veillonella*. In IDA patients compared with normal subjects, the former decreased from 2.6% to 1.2% and the latter increased from 0.4% to 2.0%. At the genus level, *Faecalibacterium* and *Veillonella* again increased to 14.9% and 3.9%, respectively, while *Collinsella* further decreased to 0%. Of course, all of the taxa that showed significant changes essential. However, in the case of the *Faecalibacterium* lineage, the composition ratio is very high and the range of change is very pervasive, so it seems clear that it plays a vital role in IDA. The LEfSe results support this importance.

Today, a significant member of the phylum Fir-micutes, the Clostridium class, and the Ruminococcaceae family, has become the most common bacterium in the gut mi-crobiome of healthy adults, which constitutes more than 5% of the total bacterial population and, in some cases, up to about 15% [52]. It is the most critical butyrate-producing bacterium in the colon. It has been considered a biomarker of human health, favoring the inflammatory process once populations decline, which correlates with inflammatory bowel disease and colon cancer [53]. Thus, this species is considered a valuable potential biomarker to help differentiate between ulcerative colitis and Crohn’s disease and a potentially active component of probiotics as a promising therapeutic strategy for various intestinal diseases [54]. According to this study’s NGS test results, this species accounted for 3.17% of the total in normal subjects but decreased to 2.65% in IDA patients and increased to 6.12% in subjects who recovered from IDA following iron supplementation. Although the NGS data did not show significant differences between groups, PCR validation experiments confirmed significant differences between IDA patients before and after iron supplementation. In a previous clinical study, a decrease in *Faecalibacterium* was identified as an important biomarker in subjects with gestational anemia compared with normal subjects, which is similar to the results of this study [55]. However, more research on *Faecalibacterium* in iron metabolism and IDA is needed. In one report, the administration of *F. prausnitzii* to germ-free mice induced an increase in ferritin, a protein that regulates intracellular iron homeostasis in the colon. This result could be linked to a mechanism involved in this phenomenon [56]. Additionally, *F. prausnitzii* is a representative butyrate-producing anti-inflammatory bacterium, and it seems necessary to pay attention to the anti-inflammatory effect of this butyrate in understanding this mechanism [57]. Iron and its homeostasis are closely related to the inflammatory response, and adaptations to the iron deficiency that ameliorate the inflammatory state underlie the most apparent link between chronic diseases such as inflammation or anemia [58]. In particular, the improvement from IDA following iron supplementation and the recovery of *Faecalibacterium* can be understood concerning hypoferremic conditions promoted by inflammation [58]. However, it is unknown whether IDA was induced before dysbiosis, which decreased *Faecalibacterium*, or if the causative link was reversed. We are also unsure whether iron supplementation led to *Faecalibacterium* recovery and IDA improvement or if the causal link was reversed. However, iron deficiency, the resulting IDA, the state of the gut microbiome associated with it, and the resulting inflammation are all closely connected in a complex relationship.

Additionally, the heme biosynthetic module was increased in IDA patients compared with normal subjects and lowered again in subjects who recovered from IDA following iron supplementation. Although this result may only be a prediction of a functional biomarker, it can be interpreted as another vital mechanism, as heme is essential in critical biochemical processes such as respiration in bacteria [59,60]. However, more in-depth research regarding heme biosynthesis and the role of *Faecalibacterium* is needed.

The most striking result of this study is that dysbiosis was seen in IDA patients compared with normal subjects, which tended to recover as they improved from IDA following iron supplementation. This result can be seen in the pattern change in the composition of taxa. However, there are still limitations regarding statistical significance in the composition of the bacterial community between groups, and further analysis using more subjects seems necessary. In the case of *Faecalibacterium*, which was particularly noteworthy, it was significantly reduced in IDA patients and recovered with iron supplementation, which shows the possibility of this taxon being applied to the diagnosis and treatment of IDA. However, these changes in microbiome status and taxonomic biomarkers can only be confirmed once until they are validated in animal disease models. In addition, it is necessary to pay attention to changes in *Collinsella* and *Veillonella*, and additional experiments should be conducted on them. Although functional biomarkers were predicted in this study, to truly understand the microbiome-related crosstalk in IDA, the analysis of the entire genome of bacteria in feces must be performed. Although this study had several limitations, these results can be used as primary data for developing microbiome-based IDA diagnosis, prevention, and treatment strategies for women of childbearing age.

## 5. Conclusions

A microbiome characterization study was conducted through 16S gene-based metagenomics on young, IDA-vulnerable women’s feces. As a result, it was shown that there was a significant difference in bacterial community characteristics in IDA patients compared with normal subjects, and intestinal dysbiosis during the disease state was restored as IDA improved. In addition, it was confirmed that *Faecalibacterium* was significantly reduced in IDA patients and significantly recovered to normal levels following iron supplementation. Although it had several limitations, this study is expected to serve as a basis for developing treatment and prevention strategies based on host–microbiome crosstalk, especially for IDA in women of childbearing age.

## Figures and Tables

**Figure 1 nutrients-15-00691-f001:**
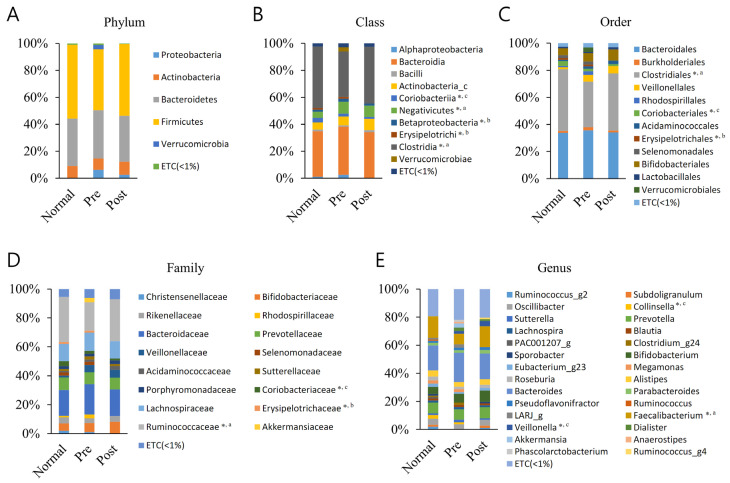
Averaged taxonomic composition of the normal group (Normal), IDA patient group (Pre), and recovery group from IDA following iron supplementation (Post). Taxonomic relative abundance was categorized at the (**A**) phylum, (**B**) class, (**C**) order, (**D**) family, and (**E**) genus levels, and relative abundances of less than 1% are expressed as ETC. The Wilcoxon rank-sum test was used to analyze the groups’ significance (* *p* < 0.05). The a, b, and c superscripts denote the groups subjected to significance determination (a, Normal–Pre; b, Pre–Post; c, Normal–Post).

**Figure 2 nutrients-15-00691-f002:**
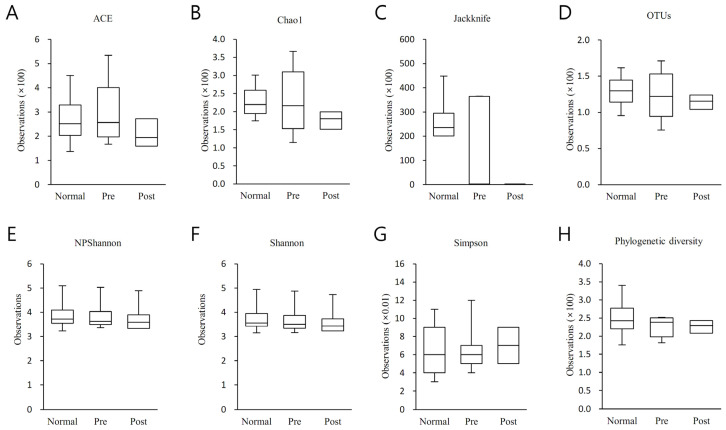
Results of the alpha diversity index analysis between normal subjects (Normal), IDA patients (Pre), and subjects who recovered from IDA with iron supplementation (Post). Species richness was analyzed using (**A**) Ace, (**B**) Chao1, (**C**) Jacknife, and (**D**) OTU, and species diversity was analyzed using (**E**) NPShannon, (**F**) Shannon, (**G**) Simpson and (**H**) Phylogenetic diversity. The analysis findings are shown as a boxplot, with the top and bottom of the box rep-resenting the first and third quartiles and the horizontal band representing the medi-an. The three groups did not differ statistically significantly from one another in any of the results analyzed above.

**Figure 3 nutrients-15-00691-f003:**
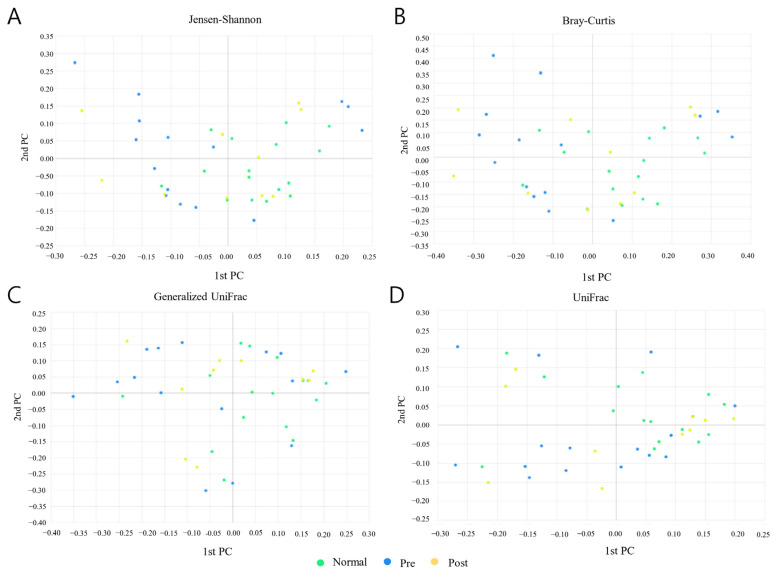
Results of PCoA (principal coordinate analysis) analysis between normal subjects (Normal), IDA patients (Pre), and subjects who recovered from IDA following iron supplementation (Post). (**A**) Jensen–Shannon, (**B**) Bray–Curtis, (**C**) Generalized UniFrac, and (**D**) UniFrac methods were applied to the analysis. Light green, blue, and yellow circles represent the Normal, Pre, and Post groups, respectively.

**Figure 4 nutrients-15-00691-f004:**
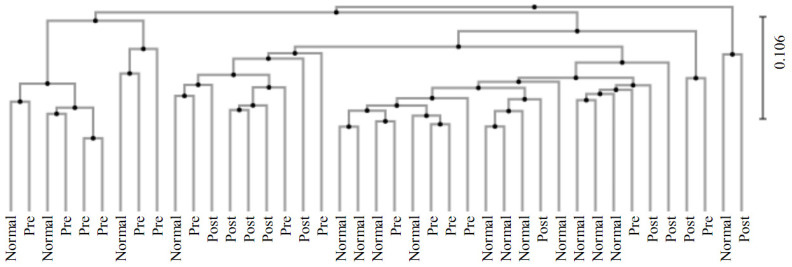
Results for clustering using the the UPGMA method between groups. The analysis of normal subjects (Normal), IDA patients (Pre), and subjects who recovered from IDA following iron supplementation (Post) was conducted.

**Figure 5 nutrients-15-00691-f005:**
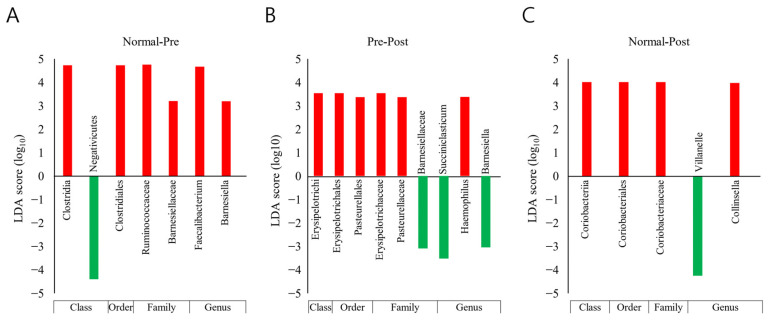
Results of taxonomic biomarker analysis between normal subjects (Normal), IDA patients (Pre), and subjects who recovered from IDA following iron supplementation (Post). The LEfSe (LDA Effect Size) analysis method was applied. Taxonomic cladograms are only presented for taxa with a significant Kruskal–Wallis test result (*p* < 0.05) and a LDA (linear discriminant analysis) score of greater than 3. Comparisons were conducted (**A**) between the Normal and Pre groups, (**B**) between the Pre and Post groups, and (**C**) between the Normal and Post groups; red denotes more abundance compared with the previously mentioned group, and green denotes the opposite.

**Figure 6 nutrients-15-00691-f006:**
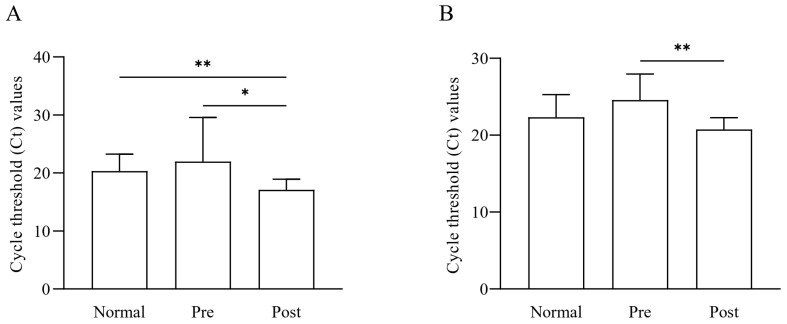
Quantification results of *Faecalibacterium* and *F. prausnitzii* among normal subjects (Normal), IDA patients (Pre), and subjects recovered from IDA following iron supplementation (Post). The bacterial loads of the (**A**) *Faecalibacterium* and (**B**) *F. prausnitzii* in the samples were quantified with qRT-PCR. Statistical analysis was performed using one-way ANOVA to determine the signifi-cance between the groups (* *p* < 0.05, ** *p* < 0.01).

**Table 1 nutrients-15-00691-t001:** The baseline characteristics of the study population.

	Normal (n = 16)	IDA-pre. Tx. (n = 15)	IDA post. Tx. (n = 10)	Normal Reference Value
Age (years)	29.25 ± 7.1	37.13 ± 7.92		
Height (cm)	161.59 ± 3.85	161.31 ± 3.48		
WBC (/µL)	6793.75 ± 1614.3	5046.67 ± 1374.7	5440 ± 1734.74	4000–10,000
Hemoglobin (g/dl)	13.43 ± 0.77	8.57 ± 1.26 *	13.31 ± 0.75	12–16
Hematocrit (%)	40.11 ± 1.93	28.11 ± 3.33 *	39.85 ± 2.54	36–48
MCV (fl)	92.52 ± 3.82	67.93 ± 7.01 *	89.19 ± 5.72	86–102
RDW (%)	14.59 ± 4.61	19.44 ± 3.45 *	13.91 ± 1.14	11.5–14.5
Platelet (10^3^/µL)	247.94 ± 45.86	344.93 ± 95.67	297.5 ± 83.64	130–450
Ferritin (ng/mL)	49.55 (32.42, 75.75)	2.31 ± 1.4 *	60.87 ± 77.28	5–204

IDA, iron-deficiency anemia; WBC, white blood cells; MCV, mean corpuscular volume; RDW, red cell distribution width; *, *p* < 0.05. Among the values corresponding to IDA patients, those marked with an asterisk were statistically significant when compared with normal subjects and simultaneously significant when compared with subjects who recovered from IDA following iron supplementation.

**Table 2 nutrients-15-00691-t002:** Average taxonomic composition of taxa that showed significant differences between groups.

Class	Norl (%)	PRE (%)	Post (%)	Order	Nor (%)	PRE (%)	Post (%)	Family	Nor (%)	PRE (%)	Post (%)	Genus	Nor (%)	PRE (%)	Post (%)
*Clostridia* *^, a^	45.6	33.8	42.2	*Clostridiales* *^, a^	45.6	33.8	42.2	*Ruminococcaceae* **^, a^	31.2	19.6	29.1	*Faecalibacterium* **^, a^	15.3	7.5	14.9
*Coriobacteriia* *^, c^	3.5	2.0	1.5	*Coriobacteriales* *^, c^	3.5	2.0	1.5	*Coriobacteriaceae* *^, c^	3.5	2.0	1.5	*Collinsella* *^, c^	2.6	1.2	0.0
*Negativicutes* *^, a^	4.5	8.8	8.4									*Veillonella* *^, c^	0.4	2.0	3.9
*Erysipelotrichi* *^, b^	1.2	1.4	0.0	*Erysipelotrichales* *^, b^	1.2	1.4	0.0	*Erysipelotrichaceae* *^, b^	1.2	1.4	0.0				
*Betaproteobacteria* *^, b^	1.5	2.2	1.5												

Taxonomic relative abundances are only presented for those with significant differences according to the Wilcoxon rank-sum test among taxa with a composition ratio of more than 1% (* *p* < 0.05; ** *p* < 0.01). Pre refers to the iron-deficiency anemia (IDA) patient group, and Post refers to the group receiving iron supplementation and recovering from IDA. The a, b, and c superscripts denote the groups subjected to difference significance determination (a, Normal–Pre; b, Pre–Post; c, Normal–Post).

**Table 3 nutrients-15-00691-t003:** Beta diversity index between normal subjects (Normal), IDA patients (Pre), and subjects who recovered from IDA following iron supplementation (Post).

Index	Normal–Pre	Normal–Post	Pre–Post
Jenson–Shannon	* (*p* = 0.037)	N.S. (*p* = 0.672)	N.S. (*p* = 1.000)
Bray–Curtis	* (*p* = 0.017)	N.S. (*p* = 0.282)	N.S. (*p* = 0.929)
Generalized UniFrac	* (*p* = 0.045)	N.S. (*p* = 0.430)	N.S. (*p* = 0.853)
UniFrac	N.S. (*p* = 0.062)	N.S. (*p* = 0.329)	N.S. (*p* = 0.594)

The beta set significance between groups was determined with PERMANOVA (Permutational Multivariate Analysis of Variance). * *p* < 0.05; N.S., non-significant.

**Table 4 nutrients-15-00691-t004:** Functional biomarker prediction results for the normal group (Normal), IDA patient group (Pre), and subject group recovered from IDA with iron supplementation (Post).

	Ortholog	Definition	LDA Effect Size	*p*-Value	Normal (%)	Pre (%)	Post (%)
Orthology	K03205	Type IV secretion system protein VirD4	2.44	0.0288	0.16	0.10	0.15
	K02315	DNA replication protein DnaC	2.36	0.0209	0.14	0.09	0.14
	K03169	DNA topoisomerase III	2.34	0.0358	0.16	0.12	0.15
	K03496	Chromosome partitioning protein	2.33	0.0098	0.21	0.17	0.21
	K06919	Putative DNA primase/helicase	2.32	0.0419	0.15	0.12	0.16
	K02316	DNA primase	2.02	0.0493	0.11	0.10	0.12
Module (PICRUSt)	M00051	Uridine monophosphate biosynthesis, glutamine (+PRPP) => UMP	2.48	0.0155	0.89	0.86	0.92
	M00121	Heme biosynthesis, glutamate => heme	2.42	0.0144	0.33	0.39	0.35
Module (MinPath)	M00854	Glycogen biosynthesis, glucose-1P => glycogen/starch	2.59	0.0096	0.33	0.24	0.32
	M00086	beta-Oxidation, acyl-CoA synthesis	2.57	0.0132	0.34	0.30	0.37
Pathway (PICRUSt)	ko04142	Lysosome	2.45	0.0124	0.29	0.35	0.29
Pathway (MinPath)	ko04910	Insulin signaling pathway	2.85	0.0110	0.34	0.49	0.39

## Data Availability

The data contained in the article and the original data that support the findings of the present study are available from the corresponding author upon reasonable request.

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
