# Peer review of "The Effects of Iron Deficiency on the Gut Microbiota in Women of Childbearing Age"

_nutrients, 2023, doi:10.3390/nu15030691_

Round 1

Reviewer 1 Report

This paper evaluates the effects of iron deficiency on the microbiota in the gut of women of childbearing age.

The authors ask the question whether there is a difference in the gut microbiota with iron deficiency anemia (IDA) and whether that change returns to normal with Fe supplementation. They conclude that the dysbiosis during the disease state was restored, and that this may become the basis for developing treatment and prevention strategies based on host-microbiome crosstalk in IDA.

They chose to study this population because other than in the USA and Europe, IDA is so prevalent in this population around the world.

The most significant results is that Faecalibacterium lineage changes the most in IDA patients and is restored in treated patients, making it likely to be an important player in IDA.

Comments:

1)      This is a nice study, but what is lacking is more about the significance of the work. The conclusions state that this study is expected to serve as a basis for developing diagnosis, treatment and prevention strategies based on host-microbiome crosstalk. This should be expanded. Where does host-microbiome crosstalk play an important role? What is the use of the microbiome in diagnosis that is better than measuring the levels of iron and related molecules in the blood? What possible treatment or prevention strategies would come from the knowledge gained in this study?

2)      The lineage that ends with Faecalbacterium has reduced levels in IDA, and returns back to normal upon supplementation. However, the lineage leading Collinsell genus while lower in IDA than in healthy women is reduced even further upon supplementation. There should be some understanding why that is so. 

3)      The data from the alpha diversity analysis showed similar diversity within each group, with no difference found among the groups in their diversity. In the beta diversity analysis, there were differences between the normal and IDA groups using almost every measure, and no difference between the normal and the post treatment groups. However, there was also no difference between the pre and post treatment groups. This should be explained in the discussion – how that relates to the findings of organism changes after patients received Fe supplementation. 

4)      There needs to an assessment of the significance of the shifts seen in the UPGMA cluster analysis. 

5)      Functional biomarkers: It is unclear what these were performed on. On the entire stool sample from each patient? The results are nice showing that changes in IDA patients compared to normal subjects are reversed after iron supplementation. 

6)      There should be some statement about the limitations of the study. 

7)      Table 1 should state what is compared with what for the statistical analysis (p values, asterisks). 

8)      Figure 1. It does not seem useful to put in so many individual bacteria, especially at the genus level as it is very difficult to make meaningful comparisons. Perhaps put the ETC at a higher level (25%? 30%?) so that the most significant information is visible. 

9)      Figure 3. The clarity is poor and the data points are very difficult to discern.

Author Response

Response to Reviewer 1 Comments

Point 1: This is a nice study, but what is lacking is more about the significance of the work. The conclusions state that this study is expected to serve as a basis for developing diagnosis, treatment and prevention strategies based on host-microbiome crosstalk. This should be expanded. Where does host-microbiome crosstalk play an important role? What is the use of the microbiome in diagnosis that is better than measuring the levels of iron and related molecules in the blood? What possible treatment or prevention strategies would come from the knowledge gained in this study?

Response 1: Thank you for pointing out the most important points in this study and giving advice. The most striking result of this study is that dysbiosis was seen in IDA patients compared to normal subjects, which tended to recover as it was improved from IDA with iron supplementation. In the case of Faecalibacterium, which was particularly noteworthy, it was greatly reduced in IDA patients and recovered while improving with iron supplementation, which is considered to show the possibility of this taxa being applied to the diagnosis and treatment of IDA. However, these changes in microbiome status and taxonomic biomarkers cannot be confirmed until they are validated in animal disease models. And although functional biomarkers were predicted in this study, in order to truly understand the microbiome-related crosstalk in IDA, analysis of the entire genome of bacteria in feces must be performed together. As suggested by the reviewer, it seems clear that the results of this study have scalability for application in the diagnosis and treatment of IDA. However, there are still some limitations that are lacking in the realization of the expansion. I have summarized these things in the last paragraph of the discussion section, so I would be very grateful if you could review them again.

<Revised manuscript line no. 460-476>

Point 2: The lineage that ends with Faecalbacterium has reduced levels in IDA, and returns back to normal upon supplementation. However, the lineage leading Collinsell genus while lower in IDA than in healthy women is reduced even further upon supplementation. There should be some understanding why that is so?

Response 2: Thank you for your careful review and comments. First of all, in the second paragraph of the discussion section, we additionally presented the results of changes in taxa for subjects who improved from IDA with iron supplementation, which was missing. Briefly misrepresenting the results, as iron supplementation improved from IDA, Faecalibacterium and Veillonella increased again to 14.9% and 3.9%, respectively, while Collinsella further decreased to 0%. In the case of changes in Faecalibacterium, which is a very important result in this study, validation experiments were also conducted and fully discussed in the discussion section, but changes in other minor taxa were not. However, like the advice given by the reviewer, it is necessary to pay attention to changes in minor taxa, and it is thought that additional research must be done on these. These are considered limitations of this study and are briefly mentioned in the last paragraph of the discussion section. Thank you again for the important advice that allowed us to have an in-depth discussion.

<Revised manuscript line no. 404-408, 460-476>

Point 3: The data from the alpha diversity analysis showed similar diversity within each group, with no difference found among the groups in their diversity. In the beta diversity analysis, there were differences between the normal and IDA groups using almost every measure, and no difference between the normal and the post treatment groups. However, there was also no difference between the pre and post treatment groups. This should be explained in the discussion – how that relates to the findings of organism changes after patients received Fe supplementation

Response 3: Thank you for your advice on an important matter. In this study, dysbiosis was observed in IDA patients compared to normal subjects, and it tended to recover as it improved from IDA with iron supplementation. By the way, in beta diversity analysis, it was statistically significant in IDA patients compared to normal subjects, but not significant between subjects before and after iron supplementation. This is probably due to the small number of subjects analyzed, and this limitation needs to be improved in future studies. And on the other hand, these results seem to be correlated with the change of Faecalibacterium, which has the largest composition ratio, but additional research is needed on this. I have summarized these in the last paragraph of the discussion section, so I would be very grateful if you could review them again.

<Revised manuscript line no. 460-476>

Point 4: There needs to an assessment of the significance of the shifts seen in the UPGMA cluster analysis

Response 4: In the UPGMA analysis, the equilibrium microbiome of subjects recovering from IDA with iron supplementation was similar to that of normal subjects, implying that they are far from the equilibrium microbiome status for IDA patients. However, this beta diversity result does not suggest statistical significance. Therefore, to compensate for this, the Beta set-significance results were shown as shown in Table 3, and the statistical significance values were presented here.

Point 5: Functional biomarkers: It is unclear what these were performed on. On the entire stool sample from each patient? The results are nice showing that changes in IDA patients compared to normal subjects are reversed after iron supplementation.

Response 5: This study is about 16S rRNA-gene based metagenomics analysis using the feces of the subjects, and from this result, the ecological changes of microorganisms at the taxa level were analyzed. In the case of this experiment, there may be limitations in analyzing functional biomarkers in that the entire genome of bacteria constituting feces was not analyzed. Nevertheless, in this study, as already mentioned in the "Materials and Methods" section, we presented functional biomarkers predicted using PICRUSt and KEGG. However, in future studies, it is necessary to analyze the entire genome of bacteria constituting feces in order to determine more accurate functional biomarkers. This is one of the limitations of this study and is briefly mentioned in the last paragraph of the discussion section.

<Revised manuscript line no. 460-476>

Point 6: There should be some statement about the limitations of the study.

Response 6: As suggested by the reviewers, there are several limitations to this study. The key results of this study, dysbiosis in IDA patients and recovery with iron supplementation, are unclear in terms of statistical significance. This is probably due to the small number of subjects analyzed, and this limitation needs to be improved in future studies. In addition, validation in disease animal models is also necessary for the expansion of Faecalibacterium determined as a taxonomic biomarker into diagnostic and therapeutic research. In addition, to truly understand the microbiome-related crosstalk in IDA, analysis of the entire genome of fecal bacteria must be performed together. These limitations are summarized in the last paragraph of the discussion part, so I would be very grateful if you could review them again.

<Revised manuscript line no. 460-476>

Point 7: Table 1 should state what is compared with what for the statistical analysis (p values, asterisks).

Response 7: Thank you for your careful review. As you pointed out, we recognized that the explanation of which group was determined by comparison with statistical significance was missing, so we added a simple sentence below the table to explain this. We would be very grateful if you could review the supplementary explanations additionally presented below Table 1.

<Revised manuscript. Table 1>

Point 8: Figure 1. It does not seem useful to put in so many individual bacteria, especially at the genus level as it is very difficult to make meaningful comparisons. Perhaps put the ETC at a higher level (25%? 30%?) so that the most significant information is visible.

Response 8: As you pointed out, Figure 1 was reviewed again, but if the ETC standard was raised higher, I am worried that even the three taxa that were determined to be significant changes could be missed when looking at the results corresponding to the genus. So please understand that we have no choice but to keep the results presented as they are. However, as the reviewer pointed out, I agree that it is somewhat complicated and difficult to understand. Therefore, I will always keep in mind the advice given by reviewer in future data analysis. Thanks again for the advice to help improve the quality of this manuscript.

Point 9: Figure 3. The clarity is poor and the data points are very difficult to discern.

Response 9: As you pointed out, I was belatedly aware of the low resolution of Figure 3 and re-inserted it with a higher quality.

Once again, I would like to sincerely thank the reviewer for reviewing this manuscript and giving valuable advice.

Edited portions of the manuscript are presented in red.

Reviewer 2 Report

Congratulations on your excellent work. Your results are very clear when you show the dynamics of the Faecalibacterium before and after the introduction of iron replacement therapy.

However, I miss a more detailed description of the therapeutic protocol in your paper. I suggest you add which iron salt you prescribed and what the therapeutic regimen was.

This information is important in determining the iron absorption capacity and thus bioavailability.

Author Response

Response to Reviewer 2 Comments

Point 1: Congratulations on your excellent work. Your results are very clear when you show the dynamics of the Faecalibacterium before and after the introduction of iron replacement therapy. However, I miss a more detailed description of the therapeutic protocol in your paper. I suggest you add which iron salt you prescribed and what the therapeutic regimen was. This information is important in determining the iron absorption capacity and thus bioavailability.

Response 1: Thank you for your kind comments. We thank you for your review of our study, and we will respond to you pointing out the lack of specificity in the information on administered iron. In relation to what you pointed out, we have provided additional detailed information about the iron administered to IDA patients in our study. It is presented in the paragraph corresponding to "Subjects Recruitment and Sample Collection" in the "Materials and Methods" section, so we would be very grateful if you could review it again.

<Revised manuscript line no. 99-100>

Modified parts of the manuscript are marked in red.
